# OpenReview forum: "BAFFLE: TOWARDS RESOLVING FEDERATED LEARNING’S DILEMMA - THWARTING BACKDOOR  AND INFERENCE ATTACKS"
_ICLR.cc/2021/Conference — Reject_

### Official Review · AnonReviewer4 · 2020-10-26
**Baffle is a new technique to protect FL model from backdoor attack through filtering and adaptive clipping and noising**

**Rating:** 6
**Confidence:** 4

**Review:**

This paper suggests a new solution to protect FL models from backdoor attacks. Under the backdoor attack, an adversary can manipulate a few clients' weight matrices to affect the final global model. In particular, the adversary wants the final model to make an incorrect prediction for certain inputs.

This paper's main idea to defend against a backdoor attack is to use clustering and adaptive clipping and noising. In the clustering phase, the aggregator uses a clustering technique to identifies the weight matrices that have been manipulated by the adversary. In the clipping and noising phase, the aggregator tries to mitigate the effect of manipulated weight matrices that could not be identified in the filtering phase.

Strengths: This paper uses an existing clustering algorithm (the HDBSCAN clustering algorithm (Campello et al., 2013)) that works best in the FL problem in identifying manipulated weight matrices. Using this clustering algorithm combined with adaptive clipping and noising, the proposed method can mitigate the backdoor attack. Moreover, BAFFLE, combined with the Secure-Two-Party Computation method, can protect the FL model from Inference attacks. The extensive numerical examples show that the proposed method outperforms other defense methods most of the time.

Weaknesses: This paper does not provide any theoretical guarantee and only applies the existing methods to mitigate the backdoor attacks. Therefore, we cannot make sure that the proposed method works in any problem or on any dataset. As the numerical examples show (e.g., see Table 2), BAFFLE cannot outperform other defense methods all the time.

Due to the lack of theoretical analysis or developing a new method/algorithm, I vote for weak acceptance (6). I ask the authors to clarify/explain the $\bf{\underline{novelty}}$ of their algorithms during the discussion period.

---

> ### Author Response · Authors · 2020-11-20
> **Reply to Reviewer 4**
>
> Thank you very much for your valuable comments! We are looking very forward to improve our paper with your guidance.
>
> Please find below our responses and a list of changes that we implemented:
>
> # Clarifications:
>
> **The novelty of our approach.**
>
> 1. A novel generic privacy-preserving FL backdoor system that simultaneously protects both the security and data privacy of FL-based applications by effectively preventing backdoor and inference attacks. To the best of our knowledge, this is the first work that discusses and tackles this dilemmatic  challenge, i.e., _no_ existing defense against backdoor attacks preserves the privacy of the clients’ data.
>
> 2. A novel generic backdoor defense (cf. Algorithm 1) that has three-folds of novelty: (1) a novel two-layer defense, (2) a new clustering approach, and (3) a new adaptive threshold tuning scheme for clipping and noising as follows:
>
>     * A novel two-layer defense concept: To the best of our knowledge, we are the first to point out that combining two classes of defenses can prevent the adversary to trade-off between attack impact and attack stealthiness. The reason is that clustering-based defenses are effective for mitigating attacks with intense manipulations (§3.1) while “clipping and noising-based” defenses are useful for eliminating weak manipulations (stealthiness) (§3.2 and §3.3). However, the naïve combination of these two classes of defenses is not effective. Our evaluation (as discussed in the last paragraph in §F.1) shows that simply stacking clustering using K-means (e.g., Shen et al., 2016) and clipping and noising (e.g., Bagdasaryan et al. 2020) is not effective to mitigate sophisticated backdoor attacks. Therefore, we introduce a new clustering approach as well as clipping and noising approaches to mitigate powerful backdoor attacks.
>
>     * A novel clustering approach tackling dynamic attack scenarios (§3.1): We utilize a completely different clustering approach (density-based clustering, HDBSCAN/DBSCAN) with appropriate parameterization so that clusters containing poisoned models are identified as outliers, irrespective of the number of backdoors (Alg. 1, Lines 6 and 7).  Existing approaches cluster model updates either into two classes using K-means (e.g., Shen et al., 2016), or only select one or a small subset of models (Blanchard et al., 2017), which is too coarse-grained and generates many false positives. Even worse, it cannot handle the simultaneous injection of multiple backdoors.
>
>     * A novel adaptive threshold tuning scheme for finding effective clipping bound (§3.2) and noise level (§3.3): Existing approaches do not discuss/consider this at all. Since Euclidean distances between the local models and the global model indicating how much the local models have changed after local training, and these distances reduce every round (cf. Fig. 3), it can be used to specify the clipping bound (Alg. 1. Lines 8 and 9) and the noise level (Alg. 1. Lines 8, 9, and 13) to adapt to the changes in each training iterations.
>
> # Changes:
>
> * We made the novelty aspects of our approach clearer in the contribution section in §1.
> Please let us kindly know of any further clarifications or changes required to clear all possibly remaining doubts and to get your support.

---

### Official Review · AnonReviewer2 · 2020-10-27
**a novel pipeline for robust FL against backdoor attack**

**Rating:** 4
**Confidence:** 3

**Review:**

This paper provides an interesting research direction for the cross-domain of federating learning and backdoor attacks. This direction has very limited work until the recent 2 years. The work being proposed in this manuscript is simple and straightforward to implement. The pipeline has been clearly demonstrated. The experiments have multiple aspects presented and show promising results in various metrics.

pros:
- Novel problem, may attract massive attention
- Simple and intuitive pipeline, conceptually easy to implement
- Results are versatile, many comparison tables are provided

cons:
- Whole article is not self-contained, feel the connections between modules are very loose
- The design of the pipeline is very ad-hoc, so many engineering aspects can be tweaked and the performance could be dramatically altered.
- The experiments could use a few popular trojan attack methods, the baselines are not comprehensive.

concerns:
- My major concern comes from the design of the pipeline and the experiments. The author(s) have created many splendid terms to describe the modules used in this work, however, their implementation uses both clustering and median, which is very engineering and may not reliable with a different clustering algorithm or data set is severely unbalanced (just like the non-iid data sets among clients). This kind of uncertainty due to the ad-hoc nature of the pipeline causes me to wonder: how bad this framework can be if any of the carefully cherry-picked modules fails its purpose? The mathematical motivation of this paper is missing and this causes the impression of untrustedness on the model design. It would be better if the author(s) can 1. provide some mathematical proofs or derivations to support your design. 2. provide a lower bound or upper bound for performance guarantee.

- Please compare it with a few Trojan attack methods in recent years. I believe no matter what kind of backdoor and Trojan attack, can be easily applied to FL by applying them individually on each client without too much trouble.
An Embarrassingly Simple Approach for Trojan Attack in Deep Neural Networks. KDD 2020

- For federated learning, one important experimental factor is the number of clients (K= 5, 10, 50, 100, etc), the portion of data on each client (1%, 5%, 10%, etc), and data distribution assumption (iid, non-iid with feature shift or label shift, etc). Please evaluate the results by changing these important hyperparameters.

- The holomorphic encryption is a pretty standard concept in FL, I don't understand why the author(s) have listed this as the major contribution for the work. Especially, only one paper from 1986 is mentioned and nothing especially has been proposed in this work. It is just an unusual way to list your contribution.

- Code is not provided, I can not see the reproducibility of this work.

minor:
please attach your main context pdf in the submission and submit the appendix in the supplementary material

---

> ### Author Response · Authors · 2020-11-20
> **Reply to Reviewer 2 (Secure two-party computation part)**
>
> # Clarifications:
>
> **Secure two-party computation (from 1986).** It seems that there is a misunderstanding here. _Nowhere_ in the paper do we mention that we would use homomorphic encryption (HE) for BAFFLE. Instead, we use secure two-party computation (STPC) techniques. STPC is more efficient than HE while also being provably secure. STPC effectively prevents the aggregator from accessing the model updates and hence mitigates inference attacks on the local models. Indeed, the pioneering work by Andrew Yao in 1986, that we cite, was ground-breaking for secure computation: Yao showed that it is possible to evaluate any efficiently computable function securely (i.e., in a privacy-preserving manner). This is why citing the paper by Yao (Yao, 1986) is standard practice in the security and privacy research community due to the role of Yao‘s work as a seminal paper in this area. However,  since Yao’s publication, an extensive line of research work followed his paradigm and introduced optimized secure computation protocols, implementations, and various efficiency improvements, e.g., point-and-permute (Beaver et al. 1990), (Kolesnikov & Schneider, 2008), FastGC (Huang et al.,2011), fixed-key AES (Bellare et al., 2013), and half-gates (Zahur et al., 2015) to name some.
> We use ABY (Demmler et al., 2015), a state-of-the-art STPC-framework that implements three techniques for secure computation: Yao’s Garbled Circuits (originally introduced by Yao in 1986), Boolean- and Arithmetic-Sharing (building upon a work by Goldreich et al. in 1987). Note, that ABY also includes recent advancements that make the original protocols from 1986/87 significantly more efficient. For details, please refer to (Demmler et al., 2015).
>
> **Our STPC contributions.** Our contribution in private BAFFLE goes significantly beyond simply applying existing STPC protocols and the ABY framework. We carefully designed all components and operations as Boolean circuits in a very efficient way. For example, we synthesized _novel_ (previously not existing) and highly optimized circuits (we will highlight this more prominently in the revised version of the paper) for the square root calculation that is also of independent interest and can be used for other applications that need a privacy-preserving computation of the square root (e.g., any protocol that uses the Euclidean distance like privacy-preserving face recognition (Osadchy et al., 2010)).
> For the circuit generation, we customized the flow of the commercial hardware logic synthesis tools (DC, 2010) to generate circuits optimized for the Garbled Circuits technique. For example, with the free-XOR  technique (Kolesnikov & Schneider, 2008; which allows to calculate all XOR operations for _free_), one has to minimize the number of non-XOR gates in the Boolean representation. We developed a technology library to guide the mapping of the logic to the circuit with no manufacturing rules defined,  similar to what was done in TinyGarble (Songhori et al., 2015). More concretely, to generate efficient Boolean circuits for private BAFFLE, we constrained the mapping to free XOR gates and non-free AND gates. We enhanced the cost functions of the single gates: We set the delay and area of XOR gates to 0, the delay and area of the inverters to 0 (as they can be replaced with XOR gates with the constant input 1), and the delay and area of AND gates to a non-0 value.
> We provide details on the protocol design of private BAFFLE’s design in §C and we will extend the details for private BAFFLE in the main part of the paper (e.g., we will elaborate more on the circuit generation).
>
> **Separating the appendix.** (“minor: please attach your main context pdf in the submission and submit the appendix in the supplementary material”)
> We surely can do this if it is required. However, we would like to point out that the structure of our paper follows closely the explicit requirements laid out in the CfP and resembles the structure of numerous other papers published in recent years at ICLR. It is therefore not quite clear to us what benefit it would provide to the reader to push materials from the appendices to supplementary materials as this would likely result in making these materials somewhat more difficult to access and therefore decrease the readability of the paper as a self-contained publication.
>
> # Changes:
>
> * We clarified our contributions when designing and implementing private BAFFLE in §4 and §D.
>
> Please let us kindly know of any further clarifications or changes required to clear all possibly remaining doubts and to get your support.

---

> ### Author Response · Authors · 2020-11-20
> **Reply to Reviewer 2 (Backdoor defense part)**
>
>
> Thank you very much for your valuable comments! We are looking very forward to improve our paper with your guidance.
>
> Please find below our responses and a list of changes that we implemented:
>
> # Clarifications:
>
> **Loose connection and ad-hoc design, unreliable clustering and median.** We do not quite follow the reviewer‘s criticism on why the design of our approach should be considered ad-hoc, as our solution seeks to cater generic requirements: we provide a design that can be generalized to different use cases and datasets, and that is not dependent on specific characteristics of the underlying data.
> We demonstrate independence of specific use cases by applying our solution to a number of quite different datasets stemming from very different application areas. It is, therefore, difficult to appreciate why the approach should be considered merely ad-hoc, as we do not see any limitations in why the  approach presented in our paper could not be extended to even further scenarios. We would therefore kindly ask the reviewer to more specifically articulate the reasoning why the approach would not be generalizable (i.e. ad-hoc) as suggested by the review.
> On the contrary, we believe the presented approach is generic: First, we introduce a generic approach consisting of a two-layered defense that can prevent an adversary to simultaneously achieve both goals defined in §2: attack impact and attack stealthiness. Second, we show that a simple combination of existing approaches is not effective (cf. the last Paragraph in §F.1). We then systematically evaluate the effectiveness of each component of BAFFLE and their combination (cf. 5.2). We extensively evaluated our approach against state-of-the-art attacks and defenses on typical FL settings introduced in top-tier machine learning and security conferences (e.g., ICLR (Xie et al., 2020), AISTATS (Bagdasaryan et al., 2020), USENIX Security (Fang et al., 2020), and IEEE S&P (Melis et al., 2019)).   We systematically designed a layered system based on a systematic requirement analysis, and we are happy to provide detailed answers to any specific and concrete concern of the reviewer.
>
> **Non-iid dataset.** We do _not_ make any assumptions about data balance or data distribution, i.e., BAFFLE is successful in mitigating backdoors in FL independent of whether the clients hold unbalanced and non-iid datasets or not. For example, in our experimental setup for the Reddit dataset, each client obtains the posts of a Reddit user. Users have different styles of writing and their posts can contain varying content. Moreover, the number of posts of each user and the size of each post vary significantly. Therefore, in our experiments with the Reddit dataset, the clients hold non-iid and unbalanced data (§E.1). For the image classification dataset, we evaluate the impact of the degree of non-iid Data (§F.1, the second paragraph). It shows that BAFFLE is effective independently of the data distribution. For the IoT dataset, each client holds a different chunk of traffic from different IoT devices.
>
> **Evaluating different trojaning strategies.** Many of the existing strategies for trojaning a model were designed for centralized training, where the adversary has full control not only over the weights but can also the model structure, which is _not_ possible in federated learning because the model structure is fixed in the FL system. Otherwise, aggregation of models from different participants would not be possible. For example, the approach by Tang et al. [2] that the reviewer mentions injects an additional sub model (TrojanNet) to the model, which can be easily detected by an aggregation server by comparing the structure of the local models to the global models. In contrast, we evaluated state-of-the-art backdooring attacks from top-tier machine learning as well as security conferences, e.g., of (Shen et al. 2016, Xie et al. 2020, Bagdasaryan et al., 2020). They consider different threat models, different kind of triggers (patch-based triggers, as proposed by Xie et al. 2020 and semantic backdoors (Shen et al. 2016, Bagdasaryan et al., 2020)) to cover different strategies for backdoor attacks that are applicable to federated learning.
>
> **Experiments with different numbers of clients.** We have run the experiments for varying numbers of clients as the reviewer suggested and no significant differences were observed (cf. §F.3)
>
> # Changes:
> * We added a summary of the discussion of non-iid datasets in the second paragraph in §5 and referred to §F.1 for the details. We added information about the distribution of data for the text prediction scenario to §E.1.
> * We added an experiment to evaluate the effectiveness of BAFFLE on different numbers of clients (ranging from 5 to 100 clients) in §F.3. It shows that BAFFLE is not affected by the number of clients.
>
> # References:
> [2] Tang, Ruixiang, et al. "An embarrassingly simple approach for trojan attack in deep neural networks." ACM SIGKDD, 2020.

---

### Official Review · AnonReviewer1 · 2020-10-28
**an interesting work**

**Rating:** 6
**Confidence:** 4

**Review:**

In the paper, the authors proposed a novel privacy-preserving defense approach BAFFLE for federated learning which could simultaneously impede backdoor and inference attacks. To impede backdoor attacks, the Model Filtering layer (i.e., by dynamic clustering) and Poison Elimination layer (i.e., by noising and clipping) were presented respectively for the malicious updates and the weak manipulations of the model. To thwart inference attacks, private BAFFLE was built to evaluate the BAFFLE algorithm under encryption using secure computation techniques.

[Strengths]

1. The paper is clear, logical, and easy to follow.
2. The topic of simultaneously defending against the backdoor and the inference attacks is significant.
3. Evaluations were conducted on multiple datasets and applications, including image classiﬁcation, word prediction, and IoT intrusion detection.

[Weaknesses]

1. The topic is significant but the contributions to the proposed approach are limited.
2. To impede backdoor attacks, many models are marked as outliers and discarded, clipped, and noised, generally speaking, which could lead to performance degradation. However, there is only a negligible effect on performance. What is the cause of this phenomenon？
3. In FL, clients locally train model updates using private data and provide these to a central aggregator. If some models were directly discarded in the central aggregator, the corresponding private data are not utilized for model training which is not an ideal approach, especially, the private data is irreplaceable.
4. Clipping and noising are the means of eliminating weak manipulations. How about the settings of them affect the results? Some analysis and ablation experiments are needed.

After reading the response, I still think that the work is promising and would like to keep my recommendation.

---

> ### Author Response · Authors · 2020-11-20
> **Reply to Reviewer 1**
>
> Thank you very much for your valuable comments! We are looking forward to improve our paper with your guidance.
>
> Please find below our responses and a list of changes that we implemented:
>
> # Clarifications:
>
> **Limited contributions.** We would kindly ask the reviewer to specify more precisely *what* the limitations referred to are. Our main contribution is to propose a _novel_ and _generic_ approach to tackle state-of-the-art backdoor and inference attacks. These attacks have been recently introduced in top-tier machine learning as well as security conferences, e.g., ICLR (Xie et al., 2020), AISTATS (Bagdasaryan et al., 2020), USENIX Security (Fang et al., 2020), and IEEE S&P (Melis et al., 2019).  Our extensive evaluation shows that our approach mitigates all these attacks as well as adaptive attacks effectively. It is common sense in security research community to show that any defenses (also generic ones) can successfully mitigate known attacks in the literature. Thus, we address the deficiency of state-of-the-art defenses in a way that to the best of our knowledge has not been proposed before. And, on top of that we also solve another challenging problem by designing these defenses in a privacy-preserving manner.
>
> **Minimum effect on the performance of the global model.** Mitigating backdoors, while retaining the performance of the model is the main goal of our paper. We aim at minimizing the following parameters:  (1) the number of models that are filtered out, (2) the clipping bound, and (3) the noise level, as all of these would negatively impact the benign performance of the model. Our dynamic clustering approach only removes models that potentially have high attack impact in order to avoid falsely rejecting benign models (cf. §3.1). We introduce adaptive clipping (§3.2) and adaptive noising (§3.3) so that the clipping bound and noise level dynamically adapt to the changes in the models in different training iterations. Moreover, since the poisoned models having high attack impact are filtered out by the Model Filtering layer, this reduces the burden to the Poison Elimination layer to mitigate backdoors, i.e., the clipping bound is increased (the model is clipped less) and noise level is decreased (less noise needs to be added). As a result, our clipping and noising approaches do not degrade the performance of the model.
>
> **Effect of the discarded models on the performance.** We agree that in the ideal case all benign models should be kept. Our approach, therefore, aims at minimizing the number of models that are falsely filtered out. By doing this, we significantly improve over state-of-the-art approaches, e.g., Krum (Blanchard et al., 2017) or Auror (Shen et al., 2016). For example, Krum only selects a single centroid local model as the global model, or collects a small subset of local models (Multi-Krum) to be aggregated in the global model, and thus, ignores contributions of a significant amount of benign models. In addition, Auror clusters model updates into two classes using K-means clustering, which is too coarse-grained and generates many false positives. Even worse, it cannot handle the simultaneous injection of multiple backdoors (cf. §3.1).
>
> **How the settings of clipping and noising affect the results.** We agree that eliminating weak manipulations is the primary goal of our clipping and noising components (the Poison Elimination layer). In §5.2, we analyzed and evaluated the effect of these components. As shown in Fig. 5, Poison Elimination is only effective if the poisoned data rate is below 13%. However, Model Filtering is effective above the poisoned data rate of 13%, as it then can reliably identify poisoned models. Moreover, we also evaluated different clipping and noising thresholds to justify our choices in Appendix F.1, Paragraph 3 and 4 and illustrated the results in Fig. 7 and 8.
>
> # Changes:
>
> * We added a summary of the discussion on clipping and noising in Sect. 5.2 and refer to Appendix F.1, Paragraph 3 and 4.
> Please let us kindly know of any further clarifications or changes required to clear all possibly remaining doubts and to secure your support.

---

### Official Review · AnonReviewer3 · 2020-10-29
**Overall the paper provides a technically sound solution for the backdoor-resilient federated learning method. But it lacks justification for the privacy enabled BAFFLE.**

**Rating:** 6
**Confidence:** 4

**Review:**

The paper proposes a backdoor-resilient federated learning method to defend the backdoor attack of poisoning the models. Their method consists of the dynamic clustering, adaptive clipping and noise adding. There are extensive experiments to demonstrate the effectiveness.

The cosine distance calculation between W_i and W_j is conducted of every pair of the models. The clustering is based on the assumption that the poisonous and benign models can be classified into two parts. In Algorithm 1, line 9, is fetching median euclidean distance a safe clipping threshold to remove the outliers? Some study is encouraged to discover such choice.

How to choose the parameter \lambda such that adding the noise N(0, sigma) do not flooding the model G_t? As training the global model G_t is with many iterations, each iteration adding certain level of noise, how to guarantee the model training convergence?

As the framework under federated learning, would the framework consider the communication cost between the global model and the local models? i.e., how many model synchronizations are needed? What are the overall operation complexity?

For Private BAFFLE, what is the consideration to choose STPC? As in experiments, there are differential privacy based method? Would the authors provide the reasoning to compare different privacy methods?

---

> ### Author Response · Authors · 2020-11-20
> **Reply to Reviewer 3**
>
> Thank you very much for your valuable comments! We are looking very forward to improve our paper with your guidance.
>
> Please find below our responses and a list of changes that we implemented:
>
> # Clarifications:
>
> **A safe clipping threshold.** Our adaptive clipping threshold (the median euclidean distance threshold) is safe and effective because: (1) We select the median Euclidean distance as a basis for clipping.  Since we assume that more than 50% of clients are benign (a common assumption in the literature), we can be sure that this value is always computed between a benign local model and the global model. (2) The Euclidean distances are reduced after each training round since the global model converges. Hence, the clipping bound is adaptively reduced to avoid damaging the performance of the global model (cf. §3.2, Fig. 3). We justify our clipping threshold choice in Appendix F.1, Paragraph 3 (Effectiveness of Clipping), in which we empirically compare our choice to the static approach as well as to other potential thresholds. Figure 7 shows that using the median Euclidean distance as the threshold can effectively mitigate backdoors while retaining the main task accuracy.
> Note: Further, it is worth clarifying that the median Euclidean distance threshold is not used to remove outliers like it is done with the clustering approach. Instead, it is used to eliminate the poisoned model weights by scaling down local models with high Euclidean distances to the global model (cf. Alg. 1, Line 11) in order to reduce attack impact. In particular, it hinders adversaries to scale up malicious model updates, e.g., as done in the model replacement-attack (Bagdasaryan et al., 2020).
>
> **Specifying $\lambda$ and noise level.** In BAFFLE, the noise level $\sigma$ is calculated based on $\lambda$ and the median of the Euclidean distances $S_t$ (cf. Alg.1, Line 13). $S_t$ weighs the noise level according to the difference between local and global models: the level of noise is dynamically reduced after each training round as the local models converge towards the global model (Bagdasaryan et al., 2020). We empirically determined $\lambda = 0.001$ for image classification and word prediction, and $ \lambda = 0.01$ for the IoT datasets.  With this, we ensure that we do not add too much noise, as this would damage the main task accuracy of the global model $G_t$, and that we still effectively mitigate backdoors in combination with our dynamic clustering and adaptive clipping approach. To justify our choice, we have run an experiment to compare the effectiveness of different $\lambda$ values and noise levels in Appendix F.1, Paragraph 4 (Effectiveness of Adding Noise), and depict the results in Fig. 8.
>
> **Overhead of BAFFLE.** We made experiments where all participants started from a randomly initialized model. In average, 1.67 additional rounds are needed. The details are discussed in appendix F.10 . Besides these small number of additional rounds, _no_ further overhead is created by BAFFLE.
>
> **STPC vs. DP for private BAFFLE.** As pointed out by the reviewer, differential privacy (DP) might be a tempting choice to reduce the information leaking from model updates. DP is a statistical approach that can be relatively efficiently implemented, however,  it can only offer effective privacy protection at the cost of a severe loss in model accuracy due to the high amount of noise that needs to be added to the models (see Zhang et al., 2020; Aono et al., 2017; So et al., 2019). This is the reason why we chose to use Secure Two-Party Computation (STPC): It guarantees strong privacy as well as high efficiency (compared to other cryptographic techniques such as homomorphic encryption [1]). Hence, STPC represents the best possible choice and trade-off to achieve provable privacy, efficiency, and accuracy. We discuss this in §4.
>
> # Changes:
>
> * We added a summary of the discussion on clipping and noising in §3.2 and §5.2 and refer to Appendix F.1, Paragraph 3 and 4.
>
> * We also added an experiment to show that our backdoor defense does not require too many more iterations (cf. Appendix F.10).
>
> Please let us kindly know of any further clarifications or changes required to clear all possibly remaining doubts and to secure your support.
>
> # References:
> [1] Gentry, Craig, and Dan Boneh. A fully homomorphic encryption scheme. Stanford University, 2009.

---

### Decision · Program_Chairs · 2021-01-07
**Final Decision**

**Decision:**

Reject

**Comment:**

The paper makes an attempt towards byzantine resilient federated learning, in the pressneece of backdoor attacks.

The method presented combines a clustering step with a poison elimination step, and seems to be effective against a range of current attacks.

Both steps are a bit ad hoc in nature, and do not come with provable guarantees.

Moreover, the algorithms presented will have a big negative impact on personalization as several models may be incorrectly discarded during and FL round.

The authors further point in their response that " no existing defense against backdoor attacks preserves the privacy of the clients’ data." This is in fact not true, as the differential privacy defense presented by the "Can you really backdoor FL" paper is in fact fully respective of user privacy.

At the same time, the work on backdoor attacks and defenses is reminiscent of the "cat and mouse" work in adversarial examples: an attack comes out, then a defense claims to protect against it, then an attack that incorporates that defense can be made stronger, and so on. This is similar in the context of backdoor attacks.

In fact, a recent work [1] proposes that detecting backdoors is in the general computationally unlikely, rendering the generality of the proposed algorithm questionable, and also suggest a set of attacks that seem very hard to defend against. (it is fine that the authors do not reference this work as it was published just recently)

As the paper lacks significant algorithmic novelty, solid guarantees, and also is unclear whether it is universally sound, the overall contribution is limited.

[1] Wang et al. Attack of the tails: Yes, you really can backdoor federated learning, neurips 2020
https://papers.nips.cc/paper/2020/file/b8ffa41d4e492f0fad2f13e29e1762eb-Paper.pdf

---

> ### Author Response · Authors · 2021-01-22
> **Reply to Final Decision and an Updated Version (part I)**
>
> We thank the reviewers for their useful comments that helped us to improve our paper. We have addressed the comments and uploaded an updated version of our paper on (https://arxiv.org/abs/2101.02281) for future readers since as all reviewers also agreed, our work is important and interesting.
> In particular, we addressed a number of critical comments as follows:
> >>“The method presented combines a clustering step with a poison elimination step, and seems to be effective against a range of current attacks.
> Both steps are a bit ad hoc in nature, and do not come with provable guarantees.”
>
> To clarify the theoretical reasons and rationale why our proposed approach mitigates known state-of-the-art backdoor attack techniques, we have added a formal theoretical argumentation to explain and justify the effectiveness of our approach (cf. Sect. III.A and III.B in https://arxiv.org/abs/2101.02281).
>
> >>“In fact, a recent work [1] proposes that detecting backdoors is in the general computationally unlikely, rendering the generality of the proposed algorithm questionable, and also suggest a set of attacks that seem very hard to defend against. (it is fine that the authors do not reference this work as it was published just recently)”
>
> Indeed, this other paper was published only very recently, and was NOT available at the time of writing. However, we took the paper and evaluated our approach against the Edge-Case attack proposed in the paper.  Our extensive evaluation shows that our approach can effectively mitigate also this novel attack. Our defense reduces backdoor accuracy in average from 42.8% to 4.0% (compared to 3.8% if no attack is launched), while preserving main task accuracy at 79.2% compared to 84.2% without defense. Thus, this confirms the general applicability of our approach. We have included the results of this evaluation in the updated version of our paper (cf. Sect. VI.A in https://arxiv.org/abs/2101.02281).
>
> We would kindly like to clarify some other comments as follows:
> >>"Moreover, the algorithms presented will have a big negative impact on personalization as several models may be incorrectly discarded during and FL round.”
>
> The main goal of FL is to train a "general/global" model rather than a "personalized" one. Personalization can, however, be done on the client side. For example, each client can improve upon the global model (i.e., personalize) by re-training the final global model with its own local data and use it immediately. For instance, Gboard, the Google Keyboard on Android, applies this approach to personalize text prediction based on what the user types on the keyboard [[2]](https://ai.googleblog.com/2017/04/federated-learning-collaborative.html). We have included this discussion in our updated paper (cf. Sect. III.C in https://arxiv.org/abs/2101.02281).
> In our paper, we focus on mitigating backdoors while maintaining the accuracy of the global model. As explained in the answer to reviewer #1, our approach can minimize the negative impact on main task accuracy. As clearly shown by the evaluation, our approach has a negligible effect on the performance of the model, and it outperforms existing approaches significantly, which is thus one of the major contributions of our approach. Our approach reduces the main task accuracy by only about 0.2% whereas previous approaches caused a significant decrease between 10% and 50%.
>
> [1] Wang et al., NeurIPS 2020, Attack of the tails: Yes, you really can backdoor federated learning, https://papers.nips.cc/paper/2020/file/b8ffa41d4e492f0fad2f13e29e1762eb-Paper.pdf
>
> [2] Mcmahan et al., Google AI 2017, Federated learning: Collaborative Machine Learning without Centralized Training Data, https://ai.googleblog.com/2017/04/federated-learning-collaborative.html

---

> > ### Author Response · Authors · 2021-01-22
> > **Reply to Final Decision and an Updated Version (part II)**
> >
> >
> > >>"At the same time, the work on backdoor attacks and defenses is reminiscent of the "cat and mouse" work in adversarial examples: an attack comes out, then a defense claims to protect against it, then an attack that incorporates that defense can be made stronger, and so on. This is similar in the context of backdoor attacks."
> >
> > We are very surprised by the logic of this argument and the conclusion drawn by the reviewer. Top security and cryptography conferences, but also recently ICLR, have several papers that present attacks and defenses on a system, and in many cases, the presented defenses can mitigate only a certain class of attacks and NOT ALL. At ICLR and other top conferences, there were subsequent attack and defense papers, e.g., [[3]](https://arxiv.org/abs/1312.6199) attack -> [[4]](https://arxiv.org/pdf/1511.04508.pdf) defense -> [[5]](https://arxiv.org/pdf/1611.01236.pdf) attack -> [[6]](https://arxiv.org/abs/1706.06083) defense -> [[7]](https://arxiv.org/pdf/1802.00420.pdf) attack -> [[8]](https://openreview.net/pdf?id=BygANhA9tQ) defense.
> >
> > In contrast, our mitigation techniques can mitigate _state-of-the-art_ backdoor attacks, and our defense is generic in the sense of the byzantine adversary model.
> >
> >
> > [3] Szegedy et al., ICLR 2014, Intriguing properties of neural networks, https://arxiv.org/abs/1312.6199, (attack).
> >
> > [4] Papernot et al., IEEE S&P 2016, Distillation as a Defense to AdversarialPerturbations against Deep Neural Networks, https://arxiv.org/pdf/1511.04508.pdf, (defense).
> >
> > [5] Kurakin et al., ICLR 2017, adversarial machine learning at scale, https://arxiv.org/pdf/1611.01236.pdf, (attack).
> >
> > [6] Madry et al., ICLR 2018, Towards Deep Learning Models Resistant to Adversarial Attacks,https://arxiv.org/abs/1706.06083, (defense).
> >
> > [7] Athalye, et al. ICML 2018, Obfuscated Gradients Give a False Sense of Security: Circumventing Defenses to Adversarial Examples, https://arxiv.org/pdf/1802.00420.pdf, (attack).
> >
> > [8] Zhang et al., ICLR 2019, COST-SENSITIVE ROBUSTNESS AGAINST ADVERSARIAL EXAMPLES, https://openreview.net/pdf?id=BygANhA9tQ, (defense).